# Polymorphisms within the *PRKG1* Gene of Gannan Yaks and Their Association with Milk Quality Characteristics

**DOI:** 10.3390/foods13121913

**Published:** 2024-06-18

**Authors:** Fen Feng, Guowu Yang, Xiaoyong Ma, Juanxiang Zhang, Chun Huang, Xiaoming Ma, Yongfu La, Ping Yan, Pingcuo Zhandui, Chunnian Liang

**Affiliations:** 1Key Laboratory of Yak Breeding of Gansu Province, Lanzhou Institute of Husbandry and Pharmaceutical Sciences, Chinese Academy of Agricultural Sciences, Lanzhou 730050, China; feng990111@163.com (F.F.); xueshengyangguowu@163.com (G.Y.); abdullah33@163.com (X.M.); 15103990593@163.com (J.Z.); johnchun825@163.com (C.H.); maxiaoming@caas.cn (X.M.); layongfu@caas.cn (Y.L.); pingyanlz@163.com (P.Y.); 2Key Laboratory of Animal Genetics and Breeding on Tibetan Plateau, Ministry of Agriculture and Rural Affairs, Lanzhou 730050, China; 3Institute of Animal Husbandry and Veterinary Medicine, Tibet Academy of Agriculture and Animal Husbandry Sciences, Lasa 850004, China; 4Plateau Agricultural Science and Technology Innovation Center, Lasa 850004, China

**Keywords:** yak, *PRKG1*, SNPs, milk quality

## Abstract

Yak milk, known as the “liquid gold”, is a nutritious food with extensive consumption. Compared with cow milk, yak milk contains higher levels of nutrients such as dry matter, milk fat, and milk protein, which demonstrates great potential for exploitation and utilization. Protein kinase cGMP-dependent 1 (*PRKG1*) is an important functional molecule in the cGMP signaling pathway, and its significant influence on milk fatty acids has been discovered. The aim of this study is to explore the correlation between single nucleotide polymorphisms (SNPs) in the *PRKG1* gene and the quality traits of Gannan yak milk in order to identify candidate molecular markers for Gannan yak breeding. In this study, genotyping was performed on 172 healthy, 4–5-year-old lactating Gannan yaks with similar body types, naturally grazed, and two to three parity. Three SNPs (g.404195C>T, g.404213C>T, and g.760138T>C) were detected in the *PRKG1* gene of Gannan yaks, which were uniformly distributed in the yak population. Linkage disequilibrium analysis was conducted, revealing complete linkage disequilibrium between g.404195C>T and g.404213C>T. After conducting a correlation analysis between SNPs in the *PRKG1* gene and milk quality in Gannan yaks, we found that *PRKG1* SNPs significantly increased the content of casein, protein, and SNFs in yak milk. Among them, the TT homozygous genotype at the *PRKG1* g.404195C>T loci exhibited higher casein and protein contents compared to the CC and CT genotypes (*p* < 0.05). The SNP g.760138T>C locus was associated with casein, protein, SNFs, and TS traits (*p* < 0.05). The CC genotype had higher casein and protein contents than the TT and TA genotypes (*p* < 0.05). However, there were no significant differences in milk fat, lactose, and acidity among the three genotypes (*p* > 0.05). In summary, *PRKG1* gene polymorphism can serve as a candidate molecular marker for improving milk quality in Gannan yaks.

## 1. Introduction

Known as the “Ship of the Plateau”, the yak is an endemic species that inhabits the Tibetan Plateau and its surrounding areas, serving as a cornerstone for maintaining ecosystem functions and developing pastoralism in this region. Its thick coat and robust cardiopulmonary capabilities enable it to withstand the harsh conditions of high altitude and low oxygen levels, allowing for exceptional adaptation to the severe environment of the Tibetan Plateau. The yak provides the Tibetan people and herders with essentials such as food, transportation, fuel, clothing, and shelter, and it is intricately linked to Tibetan culture, religion, and society [1,2]. Yak milk is a high-quality dairy product and one of the best raw materials for producing various dairy products such as yogurt, butter, cream, milk, and cheese. It holds significant economic importance for herders [3]. In addition, yak milk and its dairy products are also the foundation and main source of nutrition for the daily food intake of Tibetan herdsmen. Compared with cow milk, yak milk has a lower yield, but its protein content, fat content, and dry matter content are all higher than cow milk, and it has more comprehensive bioactive ingredients and unique microbial resources [4,5]. In recent years, with the continuous improvement of people’s living standards, the demand for high-quality dairy products has been increasing. Therefore, improving the yield and quality of yak milk has become a focus of concern for herdsmen in high-altitude areas.

The yield and quality of yak milk are influenced by various complex factors, including environmental temperature, nutrition, management measures, physiological status, and genetics. Research has found that genes such as *DGAT1*, *SCD1*, *FAT3*, *HTR1B*, *CPM*, *PRKG1*, *MINPP1*, *LIPJ*, etc. are closely related to milk production traits [6,7,8]. Shi et al. [9] have discovered through an association analysis using the whole genome sequence data of Chinese Holstein cows that *PRKG1* has a significant impact on the fatty acid composition of cow’s milk, indicating its potential as a candidate gene for improving milk quality. In addition, *PRKG1* is significantly correlated with milk production in dairy goats [10]. *PRKG1* is a cGMP-dependent protein kinase (PKG) with a length of approximately 1442 kb and contains 20 exons. It plays a key regulatory role in fat factor secretion and is closely associated with feed efficiency [11], growth traits [12], and milk component traits. In addition, *PRKG1* plays an important role in asthma [13], thoracic aortic disease [14], and cancer [15,16]. A study has found that the NO/cGMP/PKG signaling pathway alleviates spontaneous and cisplatin induced apoptosis in human ovarian cancer cells [17].

The unique highland environment has also endowed yak milk with distinctive physiological activity. Extensive research has reported on the health benefits of yak milk, including its role in enhancing tolerance to hypoxia, combating fatigue, providing antioxidant effects, and boosting immune function, which have significant potential for the development and application of yak milk [18,19,20]. However, due to the influence of geographical conditions, the milk yield of yaks is low, and most herdsmen still adopt artificial milking technology, resulting in low production efficiency and insufficient development and utilization of yak milk. Therefore, there has been widespread interest in improving the milk yield and quality of Gannan yaks by utilizing molecular marker-assisted selection technology, combined with traditional breeding methods. Single nucleotide polymorphisms (SNPs), which are widely distributed throughout the genome, constitute a crucial element of genetic diversity. They serve as vital markers in genetic research, facilitating the identification of gene loci associated with economic traits [21]. Furthermore, they enable a deeper understanding of how genetic factors influence the economic characteristics of livestock and populations, thereby laying a scientific foundation for selective breeding [22,23]. Currently, SNPs have been used to identify genes related to milk production traits in Holstein dairy cows. However, there are few studies on SNPs associated with milk quality or milk yield in yaks. Therefore, screening molecular marker loci that have significant effects on yak milk quality traits and improving effective molecular marker methods can help with the selection and crossbreeding improvement of breeds, thus enhancing the economic benefits of Gannan yaks. In addition, the *PRKG1* gene has been studied in human medicine, but it is still unclear whether it is related to the milk quality traits of Gannan yaks. Therefore, this study aims to explore the relationship between novel SNPs in the *PRKG1* gene and milk quality in Gannan yaks in order to provide important candidate molecular markers for the breeding of Gannan yaks.

## 2. Materials and Methods

### 2.1. Ethical Approval

Guided by the Animal Ethics Procedures and Guidelines of the People’s Republic of China, all yak handling strictly adhered to good animal practices. The study was approved by the Animal Administration and Ethics Committee of Lanzhou Institute of Husbandry and Pharmaceutical Sciences of the Chinese Academy of Agricultural Sciences (Permit No. 1610322020018).

### 2.2. Experimental Animals and Milk Composition Analysis

In July 2023, milk samples of Gannan yak were collected in Xiahe County, Gannan Tibetan Autonomous Prefecture, Gansu Province, China (34.99° E, 102.92° N, altitude of 3000–3800 m). The yaks used for sampling were healthy, with similar body sizes and milk production. They were grazed on natural pastures without any supplemental concentrates or roughages. Using manual milking techniques, milk samples were directly collected from the udders of 172 lactating yaks (parity 2–3), and the components of the milk, including fat, protein, lactose, casein, non-fat milk solids (SNFs), acidity, and total solids (TSs), were measured. Milk composition analysis was conducted using the MilkoScanTM FT120 milk analyzer (Foss Analytical, Hellerup, Denmark).

### 2.3. DNA Extraction

Ear tissue samples were collected from 172 Gannan yaks and stored in liquid nitrogen before being transported back to the laboratory for storage at −80 °C. DNA was extracted from the ear tissue samples following the instructions of the Magnetic Animal Tissue Genomic DNA Kit (DP341, Tiangen Biochemical Technology Co., Ltd., Beijing, China) [24]. The extracted samples were left to stand on a magnetic rack for 2 min, and after the magnetic beads were completely adsorbed, the DNA solution was transferred to a new centrifuge tube and stored under appropriate conditions. The concentration of the DNA samples was measured using a Qubit fluorometric quantitation instrument (ThermoFisher Scientific Inc., Waltham, MA, USA), and the integrity of the DNA was assessed by 1% agarose gel electrophoresis.

### 2.4. Genotyping

Genotyping was performed on 172 Gannan yaks using the Illumina Yak cGPS 7K (Illumina, Huazhi Biotechnology Co., Ltd., Changsha, China) liquid chip. Specific probes were synthesized for liquid-phase hybridization to enrich multiple target sequences at different genomic locations. Subsequently, library construction and second-generation sequencing were conducted for the target regions to obtain genotype information for all SNP/InDel marker sites. During data processing, Fastp was used to control the quality of the raw sequence data, filtering out low-quality sequences, removing sequence pairs with more than 50% of bases having a quality score of Q ≤ 20, and excluding sequences containing more than 5 N bases or with a length of less than 100. The genomic location information for SNPs was derived from the assembly results of the yak reference genome Bosgru v3.0 [25] (GCA_005887515.1).

### 2.5. Statistical Analysis

The calculation of homologous chromosomes (HOs) was conducted using the GDICALL online tool (http://www.msrcall.com/gdicall.aspx (Last accessed on 10 January 2024)). The heterozygosity (HE), effective number of alleles (NEs), polymorphism information content (PIC), and genotype and allele frequencies were calculated for the two loci. Chi-square tests and Hardy–Weinberg tests were also performed to obtain the corresponding *p*-values.

Single-factor analysis of variance (ANOVA) was conducted using IBM SPSS Statistics 25 (IBM, Armonk, NY, USA) to analyze the correlation between *PRKG1* gene polymorphisms and yak milk production traits. To analyze the influencing factors of yak milk production traits, a general linear model was employed, with appropriate simplifications made based on the current situation. The simplified model adopted Formula (1), where *Yi* represents the phenotypic value of the milk quality trait, *μ* is the population mean of the milk fat trait, *SNPi* is the fixed effect of the genotype category at the locus, and *e* is the random error effect. Differences between means were tested using Duncan’s multiple comparison test, and the results were expressed as mean ± standard deviation. The significance level was set at *p* < 0.05.
*Yi* = *µ* + *SNPi* + *e*(1)

## 3. Results

### 3.1. Analysis of PRKG1 Genotyping Results and Locus Genetic Parameters in Gannan Yak

The genotype frequencies, allele frequencies, and polymorphism information content (PIC) of the three SNPs in the *PRKG1* gene were analyzed (Table 1). The results showed that three genotypes were present at each of the three SNP loci in the Gannan yak population. Among the genotype frequencies of *PRKG1* g.404195C>T, g.404213C>T, and g.760138T>C, the CT and TT genotypes had the highest frequencies, with 0.488 and 0.389, respectively. At both g.404195C>T and g.404213C>T, the C allele frequency was 0.491, and the T allele frequency was 0.509, indicating that the mutant allele was the majority at these loci. In contrast, at g.760138T>C, the T allele frequency was 0.762, and the C allele frequency was 0.238, indicating that the unmutated allele was the majority. The PIC values for g.404195C>T, g.404213C>T, and g.760138T>C were 0.375, 0.375, and 0.297, respectively, indicating moderate polymorphism. All three SNPs in the *PRKG1* gene of Gannan yaks were in Hardy–Weinberg equilibrium (*p* > 0.05).

### 3.2. Linkage Disequilibrium Analysis of PRKG1 Gene SNPs in Gannan Yak

The linkage disequilibrium analysis of the g.404195C>T, g.404213C>T, and g.760138T>C sites in the *PRKG1* gene of Gannan yak was conducted using an online tool (https://www.bioinformatics.com.cn/ (Last accessed on 1 March 2024)) (Figure 1). The results indicated that there was a complete linkage disequilibrium between g.404195C>T and g.404213C>T, while a complete linkage equilibrium existed between g.760138T>C and each of g.404195C>T and g.404213C>T.

### 3.3. Association Analysis of Milk Traits and Genotypes of SNPs in Gannan Yak

Based on the SNP genotyping data, the correlation between each locus and yak milk composition was analyzed. The extremely strong linkage between g.404195C>T and g.404213C>T indicates that they may share common genetic characteristics. Consequently, both loci exhibited the same significance after association analysis with milk quality traits. The correlation analysis between the loci of Gannan yak and milk traits is presented in Table 2. As shown in Table 2, the g.404195C>T loci was associated with casein, protein, SNFs, and acidity traits (*p* < 0.05). The casein and protein contents of the TT homozygous genotype were significantly higher than those of the CC and CT genotypes (*p* < 0.05). The SNP g.760138T>C locus was also associated with casein, protein, SNFs, and TS traits (*p* < 0.05). The casein and protein contents of the CC genotype were higher than those of the TT and TC genotypes (*p* < 0.05). There were no significant differences in milk fat, lactose, and acidity among the three genotypes (*p* > 0.05). The results indicate that the *PRKG1* gene is mainly related to casein and protein traits in Gannan yak milk, and the homozygous mutant genotype has higher levels than the wild type (*p* < 0.05). This suggests that the mutant genotype may contribute to higher-quality yak milk.

## 4. Discussion

### 4.1. Genetic Polymorphism Analysis of the PRKG1 Gene in Gannan Yak

In recent years, with the deep integration of molecular biotechnology and traditional breeding techniques, research on improving sheep fertility at the molecular level has mainly focused on studying the phenotypic effects or correlations of major signal transduction pathways or major genes and screening molecular markers to achieve marker-assisted selection. SNP is the main type of candidate gene polymorphism, and SNP analysis has become a well-established tool for identifying associations between candidate genes and economic traits in livestock populations [26]. In this study, three SNPs were identified in the *PRKG1* gene of Gannan yak, namely g.404195C>T, g.404213C>T, and g.760138T>C. PIC, which stands for polymorphism information content, is used to measure the degree of genetic variation in a population. A higher PIC value indicates a greater number of alleles and heterozygosity, suggesting greater genetic variation and selection potential in the experimental population [27,28]. Through the analysis of the polymorphic information content at the three loci, it was found that g.404195C>T, g.404213C>T, and g.760138T>C exhibited moderate polymorphism (0.25 < PIC < 0.5), indicating a relatively high degree of variation and selection potential for these three loci in the Gannan yak population. Additionally, all three loci were in Hardy–Weinberg equilibrium, suggesting that these three SNP loci in the *PRKG1* gene possess certain genetic advantages and have basically not been influenced by factors such as mutation, selection, and genetic drift during long-term evolution and natural selection [29]. In addition, we found that the three SNPs (g.404195C>T, g.404213C>T, and g.760138T>C) in the *PRKG1* gene are all located in the intron region. Research has shown that SNPs located in intron regions do not directly alter the amino acid sequence of proteins, thus having no impact on their biological properties [30]. However, further studies have revealed that introns may affect protein formation by influencing biological processes such as splice sites, mRNA stability, or translation efficiency, which in turn can lead to changes in amino acid coding [31,32,33]. The presence of intron mutations can influence economic traits like growth characteristics [34], fat accumulation [35], and feed efficiency [36] in livestock and poultry. In summary, they can be considered as candidate loci for genetic improvement in Gannan yaks.

### 4.2. Correlation Analysis of PRKG1 Gene Polymorphism and Milk Quality Traits in Gannan Yak

Yak milk is a kind of food with high nutritional value, which is rich in proteins, fats, vitamins, and minerals. It has received much attention due to its unique nutritional components and health benefits [37]. Therefore, yak milk and its products, such as ghee, lactic acid milk, and cheese, have become an important source of nutrition for highland herdsmen. They also have a certain demand in the market, driving the development of the dairy processing industry and providing economic income for farmers [38,39]. The *PRKG1* g.404195C>T, g.404213C>T, and g.760138T>C have significant impacts on casein and protein content, and the casein and protein levels in homozygous mutant populations are higher than those in the wild type. Protein is one of the main nutrients in yak milk, not only determining its nutritional value but also influencing the texture and structure of dairy products. The proteins in yak milk are mainly divided into two categories: caseins (αS1-casein, αS2-casein, β-casein, and κ-casein) and whey proteins (α-lactalbumin, β-lactoglobulin, serum albumin, lactoferrin, and immunoglobulins) [37]. Similar to cow milk, caseins are the major proteins in yak milk, and they contain bioactive peptides [40]. Lin et al. [41,42] discovered that some casein peptides in yak milk have the ability to inhibit the activity of angiotensin converting enzyme, exhibiting an antihypertensive effect. Moreover, there are indications that these peptides may offer health benefits such as anti-inflammatory, antibacterial, and immunomodulatory activities [43,44]. Additionally, yak milk protein is reported to contain approximately 45% more essential amino acids than regular milk with glutamic acid being the most abundant amino acid in yak milk [40]. Glutamate, an important excitatory neurotransmitter, also plays a crucial role in the metabolism of sugars and fats [45]. In summary, protein and casein content are important indicators of yak milk quality. SNP mutations in the *PRKG1* gene significantly increased protein and casein content in Gannan yaks, thereby improving milk quality in Gannan yaks.

*PRKG1* was considered as one of the promising candidate gene on milk fatty acids in the Chinese Holstein population [8]. Furthermore, the *PRKG1* gene plays a lipolytic role in adipocytes by hydrolyzing triglycerides, releasing fatty acids and glycerol, thereby affecting the formation of milk fatty acids [9]. In this study, we identified three SNPs (g.404195C>T, g.404213C>T, and g.760138T>C) within the intron region of the *PRKG1* gene and discovered a significant correlation between *PRKG1* SNPs and the protein content and casein content of yak milk. Furthermore, Alim et al. [46] observed a significant correlation between SNP sites in the intron region of the Fatty Acid Synthase (*FASN*) gene and the milk fat content of Holstein cows. Similarly, the fat and protein content in milk were found to be significantly associated with five loci in the intron of the *FADS2* gene [47], aligning with the outcomes of this study. Consequently, the *PRKG1* gene emerges as a potential key candidate gene for enhancing milk quality in Gannan yaks. In the realm of dairy quality science, SNPs have been recognized as markers for a range of milk production and quality traits. In recent years, specific SNPs linked to milk protein and fat content have been pinpointed by researchers. For instance, SNPs within the *DGAT1* gene have been linked to variations in milk fat content [48], and SNPs within the *TRAPPC9* gene can function as valuable genetic markers for enhancing milk protein in cows [49]. This study found that the homozygous and heterozygous genotypes of the *PRKG1* gene mutation significantly improved the casein, protein, and SNFs of Gannan yaks by analyzing the correlation between *PRKG1* gene locus and yak milk quality traits. When the base at the *PRKG1* g760138T>C site mutates from T to C, the mutant genotype CC significantly improves the content of casein, protein, SNFs, and TS in Gannan yaks. In summary, the three SNPs of the *PRKG1* gene have had varying degrees of positive effects on the milk quality traits of Gannan yaks.

## 5. Conclusions

This study represents the initial exploration of the influence of three SNP loci within *PRKG1* on milk quality in Gannan yaks. The findings revealed a significant correlation between these three SNPs and the casein and protein content in Gannan yak milk, with homozygous mutations showing a notably higher occurrence than wild-type mutations. Consequently, the *PRKG1* gene emerges as a potential candidate gene influencing the milk quality traits of Gannan yaks, thereby laying a theoretical foundation for yak breeding.

## Figures and Tables

**Figure 1 foods-13-01913-f001:**
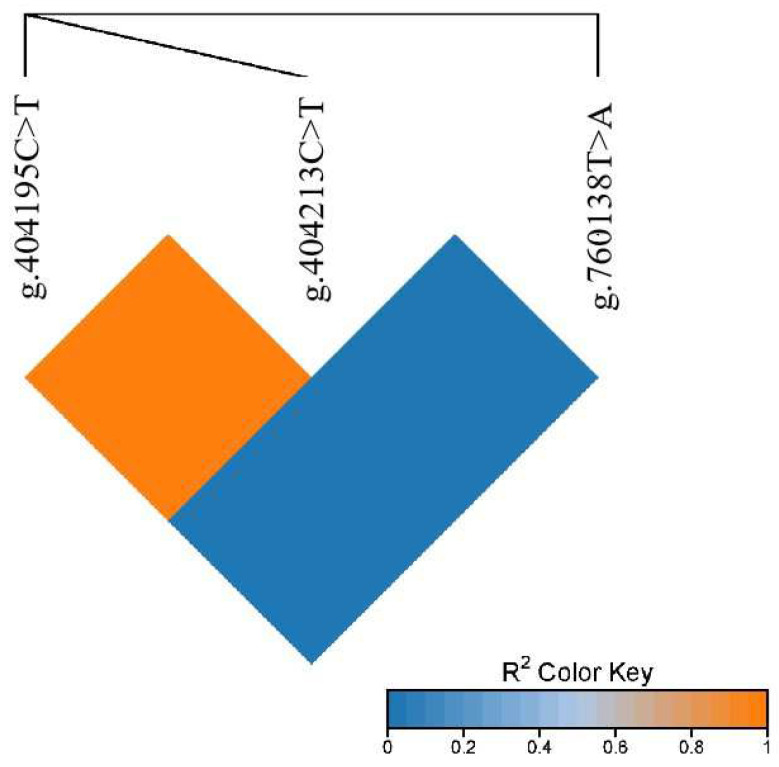
The linkage disequilibrium analysis among 3 SNPs of *PRKG1*.

**Table 1 foods-13-01913-t001:** Variation information and diversity parameters of *PRKG1* locus.

SNPs	Position	Genotypic Frequencies			Allelic Frequencies		He	Ne	PIC	*p* Value
g.404195C>T	Intron	CC	CT	TT	C	T	0.500	1.999	0.375	0.757
		0.247	0.488	0.265	0.491	0.509				
g.404213C>T	Intron	CC	CT	TT	C	T	0.500	1.999	0.375	0.757
		0.247	0.488	0.265	0.491	0.509				
g.760138T>C	Intron	TT	TC	CC	T	C	0.362	1.568	0.297	0.351
		0.568	0.389	0.044	0.762	0.238				

Note: He: heterozygosity; Ne: effective number of alleles; PIC: polymorphism. PIC < 0.25, low polymorphism; 0.25 < PIC < 0.5, moderate polymorphism; PIC > 0.5, high polymorphism; *p* > 0.05 suggests that the population gene is in the Hardy–Weinberg balance and the sample comes from the same Mendel population. CC, CT, TT and TC refer to the genotypes of SNPs in the *PRKG1* gene. C and T refer to the alleles of SNPs in the *PRKG1* gene.

**Table 2 foods-13-01913-t002:** Correlation analysis between *PRKG1* g.404195C>T, g.404213C>T, g.760138T>C, and milk traits in Gannan yak.

SNPs g.404195C>T, SNPs g.404213C>T							
Genotype	Casein %	Protein %	Fat %	SNFs %	Lactose %	Acidity °T	TSs %
CC	4.10 ± 0.25	4.91 ± 0.33	6.07 ± 2.49	11.21 ± 0.43	4.95 ± 0.13	12.36 ± 1.20	17.2 ± 2.45
CT	4.02 ± 0.31	4.81 ± 0.41	5.47 ± 2.71	11.18 ± 0.48	4.99 ± 0.16	12.18 ± 1.28	16.54 ± 2.68
TT	4.18 ± 0.25	5.04 ± 0.37	5.24 ± 2.65	11.45 ± 0.44	4.98 ± 0.15	12.79 ± 1.30	16.55 ± 2.52
*p*-Value	*p* = 0.012	*p* = 0.006	*p* = 0.34	*p* = 0.007	*p* = 0.358	*p* = 0.044	*p* = 0.383
**SNPs g.760138T>C**							
**Genotype**	**Casein %**	**Protein %**	**Fat %**	**SNFs %**	**Lactose %**	**Acidity °** **T**	**TSs %**
TT	3.97 ± 0.27	4.78 ± 0.37	4.87 ± 2.24	11.17 ± 0.41	4.99 ± 0.13	12.18 ± 1.17	15.91 ± 2.18
TC	4.11 ± 0.29	4.90 ± 0.41	5.87 ± 2.64	11.26 ± 0.48	4.97 ± 0.16	12.40 ± 1.39	17.02 ± 2.58
CC	4.25 ± 0.22	5.09 ± 0.31	6.08 ± 3.20	11.46 ± 0.51	4.98 ± 0.15	12.79 ± 1.12	17.43 ± 3.04
*p*-Value	*p* = 0.000	*p* = 0.003	*p* = 0.052	*p* = 0.037	*p* = 0.619	*p* = 0.137	*p* = 0.013

Note: SNFs: non-fat milk solids; TSs: total solids.

## Data Availability

The original contributions presented in the study are included in the article, further inquiries can be directed to the corresponding author.

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
