# Peer review of "Polymorphisms within the PRKG1 Gene of Gannan Yaks and Their Association with Milk Quality Characteristics"

_foods, 2024, doi:10.3390/foods13121913_

Round 1

Reviewer 1 Report

Comments and Suggestions for Authors

FOODS - MDPI

 Referee’s Evaluation Report: MANUSCRIPT ID: foods-2971740

Polymorphisms within the PRKG1 gene of Gannan yaks and their association with milk quality characteristics 

 (ORIGINAL ARTICLE)

Comments to Authors/Editor:

The paper of Feng & colleagues aimed to identify a possible correlation between SNPs in the Protein kinase cGMP-dependent 1 (PRKG1) gene and some quality traits of Gannan yak milk. To fulfill such objectives, a total of 172 Gannan yaks were genotyped and three SNPs (i.e., g4041…, g4042 & g.7601..) were detected in the PRKG1 gene; later correlation analyses proved an interesting correlation of these SNPs with the yak milk quality. This manuscript falls within the scope of FOODS, while it is sufficiently informative for the replication of the study.  In general, the organization of the experiment seems to be well designed, yet the English quality, grammar, and sentence structure must be improved. The Abstract was written in a careless fashion; please define SNPs (L19); interestingly, the authors defined such acronym until page 6, L211, in the Discussion section!!!; is this writing strategy scientifically sound??? Besides, please refer to L25, L27 & L29, and check this out: … SNPs significantly increased the content of …, or, SNPs increased (p< 0.05 or 0.01) the content of…. Moreover, please use only the probability lever or the word significantly, the use of both is a pleonasm; correct accordingly the whole manuscript.  In addition, the authors must inform in the Abstract section the type of yak breeds (if any) involved in the study or at least the yak scientific name, along with the age, the live weight, physiological state, and other information regarding both the experimental units as well as the site and the environmental conditions when the samples were collected. Were the three prevalent SNPs homogenously distributed along with the sampled yak population? The authors must clarify these issues for the benefit of the readers of FOODS. The authors MUST clarify some numbers, especially to include the number of animals evaluated as related to a specific geographical origin as well as the location of the study (i.e., NL, WL), how the yak was sampled regarding the collected tissues and the milk?. Why was the study carried out in Yaks and not in sheep, goats, or cows???; please contextualize.  These issues must be carefully addressed-clarified in the Abstract section. The Introduction section is generally well presented; tL47, regular milk???, what is “regular” for the authors? human milk, cow, sheep, or goat milk???; please be specific.  Prior the second paragraph, the authors must contextualize not only the importance of Asia regarding the world yak population, but also another key issues from a social, economic, and productive point of view. While the objectives of the study were included in the Introduction section, the working hypothesis of the study was never proposed; this is a must. In the Material & Methods section, I do strongly recommend starting with the general information; besides the institutional approval of the study, the authors must state if the study follows any international guide for the use of animals in research; this is a must. Section 2.2, L87, “A total of 172 milk samples….; please define the numbers of animals used to collect such 172 milk samples. In general, reagents, standards, and methods used are relevant and in accordance with the objectives of the study. Also, all the sampling techniques, laboratory methods, as well as response variables considered in the experiment are detailed and accurate, while in agreement with the objectives of the study. Nevertheless, I strongly recommend including a figure with the actual experimental protocol across time (i.e., a timeline of actions); this is a must. Section 2.5, Statistical analysis or Statistical analyses???? The experimental design was not explained, neither most of the statistical models were described for the reader to understand how the experiment was carried out. Certainly, only one statistical model was described L123-126. In the Results section, L135; … had the highest?; L157, … the same significance after…??; please include the p-value. Again, L159-160; L163-164; & L164-165 ; it is a pleonasm to include both the p-value and the word “significantly; correct accordingly the whole manuscript. L167, … has significantly higher levels; remove the word “significantly” and add the p-value instead. Are the authors comfortable with this grammar structure??? Table 2; homogenate the use of only 2 digits after the point (i.e., Genotype TT & Casein %, 0.253 or 0.25????? In general, the novelty value of the results is reasonable. In this section the authors presented their main research outcomes in diverse Figures and Tables. Nonetheless, the titles must be rewritten; certainly, the titles of tables must stand by themselves.  As extensively suggested, in this section the authors must avoid the use of both the word “significant” and the “probability value itself”; it is a pleonasm. Regarding to the Discussion section, I think the way the authors used to initiate this section, was not the best strategy to do. The opening of the Discussion with a quite old, not trendy or novelty approach, neither is the best way to attract the attention of the reader, nor to discuss the obtained results. Therefore, at the beginning of the Discussion, I do strongly suggest initiate this section including the working hypothesis of the study. Authors must define if, with the obtained results, such hypothesis is rejected or non-rejected. For this reason, the authors must include the working hypothesis prior to the objectives in the Introduction section. After this opening paragraph, the authors must follow the same order in this section according to that proposed in the Results section. Moreover, the use of subheadings in the Discussion section, following the same order as in the Results section, is highly recommended. The authors must link, in a logical fashion, their main findings along with the discussion section, to compare & to discuss and, thereafter, be able to propose some possible explanations for such specific outcomes, considering to previous similar studies from the scientific literature. Please move L211-216 to the introduction section and use it in the rationale, prior the hypothesis and aims of the study. In general, the authors made an accurate interpretation of the main findings. The authors must focus their main findings and confront them with respect to the scientific literature in a logical and focused fashion.  In general, the main outcomes of the study were not soundly presented; they were outlined in a fragile fashion. The use of subheadings will certainly help to solve such fragility.  The Conclusion section looks OK to me. The list of references cited in the manuscript is proper. This is an interesting study. Yet, the authors must improve the clarity and logical arrangement of the observed results, especially in the Discussion section.  The authors must align the conclusions regarding the working hypothesis as well as the scientific question they try to solve; nothing else, just that. All the commented issues and requests should be clearly addressed by the authors. At this point, and based on the above comments, my pronouncement is that this manuscript cannot be accepted in its actual format; it requires major adjustments. 

Comments on the Quality of English Language

The manuscript requires moderate/minor English editing.

Author Response

Reviewer 1

The paper of Feng & colleagues aimed to identify a possible correlation between SNPs in the Protein kinase cGMP-dependent 1 (PRKG1) gene and some quality traits of Gannan yak milk. To fulfill such objectives, a total of 172 Gannan yaks were genotyped and three SNPs (i.e., g4041…, g4042 & g.7601..) were detected in the PRKG1 gene; later correlation analyses proved an interesting correlation of these SNPs with the yak milk quality. This manuscript falls within the scope of FOODS, while it is sufficiently informative for the replication of the study. In general, the organization of the experiment seems to be well designed, yet the English quality, grammar, and sentence structure must be improved.

Reply: Thanks for your comments. We have revised the manuscript point to point according to your comments and suggestions.

The Abstract was written in a careless fashion; please define SNPs (L19); interestingly, the authors defined such acronym until page 6, L211, in the Discussion section!!!; is this writing strategy scientifically sound???

Reply: Thanks for your comments. We have revised it in Line 22 according to your suggestion.

Besides, please refer to L25, L27 & L29, and check this out: … SNPs significantly increased the content of …, or, SNPs increased (p< 0.05 or 0.01) the content of…. Moreover, please use only the probability lever or the word significantly, the use of both is a pleonasm; correct accordingly the whole manuscript.

Reply: Thanks for your comments. We have revised it in the whole manuscript according to your suggestion.

In addition, the authors must inform in the Abstract section the type of yak breeds (if any) involved in the study or at least the yak scientific name, along with the age, the live weight, physiological state, and other information regarding both the experimental units as well as the site and the environmental conditions when the samples were collected. Were the three prevalent SNPs homogenously distributed along with the sampled yak population? 

Reply: Thanks for your comments. We have stated it in Lines 24-26 according to your suggestion.

The authors must clarify these issues for the benefit of the readers of FOODS. The authors MUST clarify some numbers, especially to include the number of animals evaluated as related to a specific geographical origin as well as the location of the study (i.e., NL, WL), how the yak was sampled regarding the collected tissues and the milk?

Reply: Thanks for your comments. The 172 yaks in this study all originate from Xiahe County, Gannan Tibetan Autonomous Prefecture, Gansu Province, China, and are grazed in the same natural pasture. We have specified the exact locations in the 2.2 section, including any relevant geographical coordinates or landmarks. Regarding the collection of yak tissue and milk samples, we employ standardized procedures to ensure consistency and comparability among all animals. For tissue sampling, we obtain ear tissue samples from each yak's ear, preserve them in liquid nitrogen, and transport them back to the laboratory for storage at -80°C. For milk collection, we use manual milking techniques to directly gather milk samples from lactating female yaks' udders. We have included a detailed description of the sampling methods in the Line of the revised manuscript.

Why was the study carried out in Yaks and not in sheep, goats, or cows???; please contextualize. These issues must be carefully addressed-clarified in the Abstract section.

Reply: Thanks for your comments. We have clarified it in Lines 17-19 according to your suggestion.

The Introduction section is generally well presented; tL47, regular milk???, what is “regular” for the authors? human milk, cow, sheep, or goat milk???; please be specific. 

Reply: Thanks for your comments. We have revised it in Line 57 and Line 59.

Prior the second paragraph, the authors must contextualize not only the importance of Asia regarding the world yak population, but also another key issues from a social, economic, and productive point of view. While the objectives of the study were included in the Introduction section, the working hypothesis of the study was never proposed; this is a must.

Reply: Thanks for your comments. We have rewritten it which showed in Lines 82-97.

In the Material & Methods section, I do strongly recommend starting with the general information; besides the institutional approval of the study, the authors must state if the study follows any international guide for the use of animals in research; this is a must.

Reply: Thanks for your comments. We have stated it in Lines 112-116 according to your suggestion.

Section 2.2, L87, “A total of 172 milk samples….; please define the numbers of animals used to collect such 172 milk samples.

Reply: Thanks for your comments. We have defined it in Lines 122-123 according to your suggestion.

In general, reagents, standards, and methods used are relevant and in accordance with the objectives of the study. Also, all the sampling techniques, laboratory methods, as well as response variables considered in the experiment are detailed and accurate, while in agreement with the objectives of the study. Nevertheless, I strongly recommend including a figure with the actual experimental protocol across time (i.e., a timeline of actions); this is a must.

Reply: Thanks for your suggestions. At around 7 a.m., we arrived at the ranch punctually, conducted grouping, and commenced the milk sample collection process from 172 yaks.. The process was smoothly completed before 9 a.m.

Section 2.5, Statistical analysis or Statistical analyses???? The experimental design was not explained, neither most of the statistical models were described for the reader to understand how the experiment was carried out. Certainly, only one statistical model was described L123-126.

Reply: Thanks for your suggestions. In Section 2.5, we employed statistical analysis. In this study, 172 yaks were genotyped using a 7k liquid-phase chip to screen for SNP loci. To analyze the influencing factors of yak milk production traits, a general linear model was adopted for association analysis, with appropriate simplifications made based on the current situation. The simplified model is presented on Line 174, where Yi represents the phenotypic value of the milk quality trait, μ is the population mean of the milk fat trait, SNPi is the fixed effect of the genotype category at the locus, and e is the random error effect.

In the Results section, L135; … had the highest?; L157, … the same significance after…??; please include the p-value. Again, L159-160; L163-164; & L164-165 ; it is a pleonasm to include both the p-value and the word “significantly; correct accordingly the whole manuscript.

Reply: Thanks for your comments. We have revised it in the whole manuscript according to your suggestion.

 L167, … has significantly higher levels; remove the word “significantly” and add the p-value instead.

Reply: Thanks for your comments. We have removed the word “significantly” and add the p-value instead according to your suggestion, which showed in Line 220.

Are the authors comfortable with this grammar structure??? Table 2; homogenate the use of only 2 digits after the point (i.e., Genotype TT & Casein %, 0.253 or 0.25????? In general, the novelty value of the results is reasonable.

Reply: Thanks for your suggestion. We have revised it in Table 2.

In this section the authors presented their main research outcomes in diverse Figures and Tables. Nonetheless, the titles must be rewritten; certainly, the titles of tables must stand by themselves.

Reply: Thanks for your comments. We have rewritten the the titles of tables and figure according to your suggestion.

As extensively suggested, in this section the authors must avoid the use of both the word “significant” and the “probability value itself”; it is a pleonasm.

Reply: Thanks for your comments. We have revised it in the result section.

Regarding to the Discussion section, I think the way the authors used to initiate this section, was not the best strategy to do. The opening of the Discussion with a quite old, not trendy or novelty approach, neither is the best way to attract the attention of the reader, nor to discuss the obtained results. Therefore, at the beginning of the Discussion, I do strongly suggest initiate this section including the working hypothesis of the study. Authors must define if, with the obtained results, such hypothesis is rejected or non-rejected. For this reason, the authors must include the working hypothesis prior to the objectives in the Introduction section. After this opening paragraph, the authors must follow the same order in this section according to that proposed in the Results section. Moreover, the use of subheadings in the Discussion section, following the same order as in the Results section, is highly recommended. The authors must link, in a logical fashion, their main findings along with the discussion section, to compare & to discuss and, thereafter, be able to propose some possible explanations for such specific outcomes, considering to previous similar studies from the scientific literature.

Reply: Thanks for your comments. We have rewritten the discussion according to your suggestion.

Please move L211-216 to the introduction section and use it in the rationale, prior the hypothesis and aims of the study. In general, the authors made an accurate interpretation of the main findings. The authors must focus their main findings and confront them with respect to the scientific literature in a logical and focused fashion. In general, the main outcomes of the study were not soundly presented; they were outlined in a fragile fashion. The use of subheadings will certainly help to solve such fragility. The Conclusion section looks OK to me. The list of references cited in the manuscript is proper. 

Reply: Thanks for your comments. We have moved L211-216 to the introduction section, which showed in Lines 87-92.

This is an interesting study. Yet, the authors must improve the clarity and logical arrangement of the observed results, especially in the Discussion section. The authors must align the conclusions regarding the working hypothesis as well as the scientific question they try to solve; nothing else, just that. All the commented issues and requests should be clearly addressed by the authors. At this point, and based on the above comments, my pronouncement is that this manuscript cannot be accepted in its actual format; it requires major adjustments.

The manuscript requires moderate/minor English editing.

Reply: Thanks for your comment. We have carefully read the whole text and corrected the grammar and polished the sentences with the help of experts who are proficient in English.

Reviewer 2 Report

Comments and Suggestions for Authors

The study is important because it brings new knowledge about the Tibetan yak. From a formal point of view, it is well-elaborated, but I have some suggestions and comments.

The manuscript would benefit from a more detailed description of the 172 animals used for the analysis. Something more needs to be explained about farms, pedigrees (were the animals related?), management, the phase of lactation, and other factors. These could affect the milk content.

The authors analyse SNP polymorphism in gene PRKG1. For the study's comprehension, it would be beneficial if the authors better explained the steps from the gene to the new SNP. Could the authors outline the logical link between the suggestion in the introduction L74-L75 “novel SNPs in PRKG1 gene” to section 2.5. Statitical analysis L117 “calculated for the two loci" and L131 “for the three SNPs”? Could you explain two loci for one gene? Why only three SNP?

If the authors found that two SNPs have 100% linkage, why did they not supplement this study with another SNP with no linkage?

Please edit the following sentence:

L241-L242 When the base at PRKG1 241 g760138T>C site mutates from T to A, the mutant genotype TT significantly improves the content of casein, protein, SNF, and TS in Gannan yaks.

Probably, the sentence should be:

When the base at PRKG1 241 g760138T>C site mutates from T to C, the mutant genotype CC significantly improves the content of casein, protein, SNF, and TS in Gannan yaks.

Author Response

Reviewer 2

The study is important because it brings new knowledge about the Tibetan yak. From a formal point of view, it is well-elaborated, but I have some suggestions and comments.

Reply: Thanks for your comments. We have revised the manuscript point to point according to your comments and suggestions.

The manuscript would benefit from a more detailed description of the 172 animals used for the analysis. Something more needs to be explained about farms, pedigrees (were the animals related?), management, the phase of lactation, and other factors. These could affect the milk content.

Reply: Thanks for your comments. We have stated it in Lines 118-125 according to your suggestion.

The authors analyse SNP polymorphism in gene PRKG1. For the study's comprehension, it would be beneficial if the authors better explained the steps from the gene to the new SNP. Could the authors outline the logical link between the suggestion in the introduction L74-L75 “novel SNPs in PRKG1 gene” to section 2.5.

Reply: Thanks for your comments. This study started from the PRKG1 gene and ultimately identified and analyzed new SNPs in this gene through a series of steps, including genotyping, SNP screening, and association analysis. These steps together constitute a logical chain from gene to new SNP analysis, providing significant clues for us to deeply understand the role of the PRKG1 gene in specific biological processes.

Statitical analysis L117 “calculated for the two loci" and L131 “for the three SNPs”? Could you explain two loci for one gene? Why only three SNP?

Reply: Thanks for your comments. Two loci refer to the two alleles in each SNPs locus, such as the C and T in g. 404195C>T. Three SNPs refer to the three SNPs in the PRKG1 gene, such as g. 404195, g. 404213 and g.760138

If the authors found that two SNPs have 100% linkage, why did they not supplement this study with another SNP with no linkage?

Reply: Thanks for your comments. In this study, 172 yaks were genotyped through the 7k liquid-phase chip, and SNP loci were screened. The analysis revealed that three SNP loci in the PRKG1 gene were significantly correlated with yak milk quality traits, and two of them exhibited a strong linkage relationship. We did not supplement an SNP that was not in linkage in this study, primarily because we took into account the resource limitations. The 7k chip used in this study resulted in fewer loci being screened, and some loci were not significantly correlated with yak milk quality traits. However, in milk quality-related research, the PRKG1 gene has been found to be a star gene, worthy of further exploration in studies related to yak milk quality traits.

Please edit the following sentence:

L241-L242 When the base at PRKG1 241 g760138T>C site mutates from T to A, the mutant genotype TT significantly improves the content of casein, protein, SNF, and TS in Gannan yaks.

Probably, the sentence should be:

When the base at PRKG1 241 g760138T>C site mutates from T to C, the mutant genotype CC significantly improves the content of casein, protein, SNF, and TS in Gannan yaks.

Reply: Thanks for your comments. We have revised it in Line 396 according to your suggestion.

Reviewer 3 Report

Comments and Suggestions for Authors

Please take note of the following requests: - The tables require adjustments to make them self-explanatory. It would be helpful to insert in the footer of Table 1 the definitions of "CC", "CT", "TT", "C" and "T". Similarly, in Table 2, please provide the meaning of "SNF" and "TS". - In Table 2, kindly remove the "/" symbol from the column titles, so that they read as Casein %, Protein % and so on.

Author Response

Reviewer 3

Please take note of the following requests:

The tables require adjustments to make them self-explanatory. It would be helpful to insert in the footer of Table 1 the definitions of "CC", "CT", "TT", "C" and "T". Similarly, in Table 2, please provide the meaning of "SNF" and "TS". - In Table 2, kindly remove the "/" symbol from the column titles, so that they read as Casein %, Protein % and so on.

Reply: Thanks for your comments. We have added the definitions of "CC", "CT", "TT", "C" and "T" in Lines 193-194, and removed the "/" symbol from the column titles in Table2 according to your suggestion.
